# Venous Thromboembolism Risk Assessment and Prophylaxis in Trauma Patients

**DOI:** 10.3390/ijerph23010059

**Published:** 2025-12-31

**Authors:** Parichat Tanmit, Patharat Singthong, Phati Angkasith, Panu Teeratakulpisarn, Narongchai Wongkonkitsin, Supatcha Prasertcharoensuk, Chaiyut Thanapaisal

**Affiliations:** Department of Surgery, Faculty of Medicine, Khon Kaen University, Khon Kaen 40002, Thailand; paricta@kku.ac.th (P.T.); patharat.stang@gmail.com (P.S.); panute@kku.ac.th (P.T.); narwon@kku.ac.th (N.W.);

**Keywords:** deep vein thrombosis, pulmonary embolism, greenfield risk assessment profile score, thromboprophylaxis for trauma, traumatic brain injury, intracranial hemorrhage

## Abstract

**Highlights:**

**Public health relevance—How does this work relate to a public health issue?**
Venous thromboembolism (VTE) constitutes a significant cause of preventable morbidity and mortality among trauma patients globally.Standardized risk assessment scales are essential tools for identifying high-risk trauma patients who require VTE prophylaxis.

**Public health significance—Why is this work of significance to public health?**
This study demonstrates that using specific risk assessment scales can reduce the incidence of deep vein thrombosis and pulmonary embolism in trauma care settings.Preventing hospital-acquired VTE in trauma patients significantly reduces healthcare expenses and long-term complications.

**Public health implications—What are the key implications or messages for practitioners, policy makers and/or researchers in public health?**
Practitioners are advised to consistently utilize validated risk assessment scales to inform clinical decision-making concerning VTE prophylaxis within trauma departments.Hospitals and policymakers should require VTE risk stratification protocols as a standard quality indicator for trauma care.

**Abstract:**

Background: Venous thromboembolism (VTE) is associated with high morbidity and mortality. The activation of multiple pathways of venous thrombosis occurs after an injury. A prophylaxis protocol is necessary to prevent early and late venous thrombotic complications. Methods: This study aimed to evaluate the outcomes of venous thromboembolism using the Greenfield risk assessment profile score and its association with bleeding complications. This was a retrospective cohort study conducted on trauma cases who were aged 15 years or older. The study was conducted from January 2020 through December 2022. Patients who were admitted to hospital for less than 24 h or those who died during resuscitation or treatment in an emergency room were excluded from this study. Results: We enrolled 580 cases. Among them, 46.6% were categorized as high-risk for developing venous thromboembolism, and 30.4% of these high-risk patients received pharmaco-mechanical thromboprophylaxis. All VTE cases were high risk according to the Greenfield risk assessment profile, accounting for 3% of the entire group and 1.4% of all enrolled cases. All major bleeding complications occurred with a previously diagnosed large subdural hematoma. Conclusions: Assessing VTE risk was crucial for optimal management of prophylaxis. Proper use of pharmacological prophylaxis had to be balanced against the risk of bleeding complications.

## 1. Introduction

Venous thromboembolism presenting clinically as deep vein thrombosis (DVT) or pulmonary embolism (PE) is a significant cause of morbidity and mortality after traumatic injury [1,2]. The response to trauma could lead to an increased risk of thrombosis, which is caused by three factors. These factors include vascular endothelium damage from direct injury, stasis of blood flow due to hospitalization and immobilization, and hypercoagulability resulting from increased generation of thrombin and fibrin, as well as decreased levels of endogenous anticoagulants [3]. Major trauma, lower-limb fractures, and spinal cord injury are strong provoking factors for VTE [4].

Acute PE disrupts blood flow and gas exchange, leading to right ventricular (RV) failure due to acute pressure overload. Obstructive shock and hemodynamic instability result from either impaired RV filling or reduced RV flow output [1,5].

Lower extremity venous thrombosis could lead to long-term sequelae such as chronic venous outflow obstruction and secondary venous valve damage. This might result in venous obstruction, reflux, or a combination of both, ultimately causing post-thrombotic syndrome. Pulmonary embolism also has a significant impact on lung function, leading to limitations and potentially causing chronic thromboembolic pulmonary hypertension [6].

At present, the prophylaxis of venous thromboembolism involves the use of pharmacologic agents, including anticoagulants, as well as mechanical methods such as sequential compression devices (SCDs). It was reported that severely traumatized patients who did not receive pharmacologic prophylaxis had a 50% incidence of deep vein thrombosis (DVT) [7]. However, determining the optimal thromboprophylaxis was a challenge. The most effective approach to preventing venous thromboembolism involved a combination of pharmacological and mechanical (pharmaco-mechanical) methods of prophylaxis [8]. Nevertheless, this approach also elevated the risk of hemorrhage.

Recently, the Western Trauma Association released guidelines recommending the use of the Greenfield risk assessment profiles scoring system to calculate VTE risk [9].

The Greenfield Risk Assessment Profile (RAP), developed by Greenfield et al. in 1997 [10], is a validated tool for assessing trauma patients. The RAP scoring system incorporates multiple weighted risk factors across four domains: underlying medical conditions, iatrogenic factors, injury-related characteristics, and patient age.

Underlying medical conditions contributing to the RAP score include obesity (body mass index >30 kg/m^2^; 2 points), active malignancy (2 points), abnormal coagulation studies at admission (2 points), and a documented history of prior thromboembolism (3 points).

Iatrogenic factors include femoral central venous catheterization that lasts more than 24 h (2 points), massive transfusion defined as four or more blood product units within 24 h (2 points), surgical procedures that last more than 2 h (2 points), and major vascular injury requiring repair or ligation (3 points).

Injury severity is quantified using the Abbreviated Injury Scale (AIS). Regional injuries are scored 2 points each when AIS exceeds 2 in the chest, abdominal, or head regions. Severe traumatic brain injury, defined as a Glasgow Coma Scale (GCS) score less than 8 persisting beyond 4 h, contributes 3 points. Orthopedic injuries are scored as follows: complex lower extremity fractures (4 points), pelvic fractures (4 points), and spinal cord injury resulting in paraplegia or quadriplegia (4 points).

Age-stratified risk is assigned 2 points for patients aged 40–59 years, 3 points for ages 60–74 years, and 4 points for patients 75 years or older.

Total RAP scores are calculated by summing all applicable points, with individual patients potentially accumulating points from multiple factors within each domain. According to institutional protocols and published literature [10], a RAP score of 5 or higher identifies the high-risk group and indicates that pharmacologic prophylaxis could be considered when there are no bleeding contraindications.

We examined the Greenfield risk assessment profile score, VTE prophylaxis methods, VTE outcomes, and bleeding complications.

## 2. Materials and Methods

### 2.1. Objective

This study aimed to evaluate the effectiveness of our institute’s VTE prophylaxis protocol using the Greenfield risk assessment profile score.

### 2.2. Study Design and Measurements

This was a retrospective cohort study that included patients over the age of 15 who had suffered from trauma and were admitted to an academic-level I trauma center during the period from 2020 to 2022. Patients who had been admitted for less than 24 h or who died during resuscitation or treatment at an emergency room were excluded from this study.

The sample size was based on the previous prevalence of VTE in traumatic patients [11]. It was calculated based on an expected VTE prevalence of 5% in the Thai trauma population, using the formula for a single-proportion study: *n* = Z^2^ × *p*(1 − *p*)/d^2^, where Z = 1.96, *p* = 0.05, and d = 0.02.

The minimum required sample size was 207 patients. However, to ensure adequate power for subgroup analyses and to account for potential exclusions, all eligible trauma patients during the study period (*N* = 580) were included, which exceeded the minimum requirement and increased the study’s precision.

### 2.3. VTE Prophylaxis Protocol

All patients with traumatic injuries who stayed in the hospital for more than 24 h were checked for the risk of developing VTE. If a patient’s Greenfield risk assessment profile score was 5 or higher, they were classified as high risk and considered for both pharmacologic prophylaxis and intermittent pneumatic compression devices. Patients with a low-risk profile who were confined to bed also received this treatment. However, if the patient had active bleeding, coagulopathy, hemodynamic instability, solid organ injury, traumatic brain injury, or spinal injury, the administration of anticoagulants was delayed. 

Mechanical prophylaxis was administered to all bed-bound patients using bilateral sequential compression devices, which were employed for 18–20 h daily until the patients became ambulatory or were discharged. The devices were temporarily removed solely for ambulation, physical therapy, bathing, or patient comfort, with a maximum duration of 2 h.

The SCDs delivered graded compression (45 mmHg at the ankle, decreasing proximally). Active DVT in the limb was a contraindication. The pressure, cycle, and duration of compression were adjusted for patients with peripheral arterial disease. The device was modified and applied only to areas not affected by acute fractures or severe open wounds.

Pharmacologic prophylaxis included enoxaparin 40 mg subcutaneously once daily, which was reduced to 30 mg if creatinine clearance (CrCl) was below 30 mL/min. In patients with severe renal failure (CrCl < 15), when enoxaparin was contraindicated, unfractionated heparin 5000 units was administered subcutaneously every 8 h. Alternatively, patients could have continued their home anticoagulants for other reasons.

Prophylaxis was initiated when all of the following criteria were met:-No active bleeding (stable hemoglobin for >12 h, no ongoing transfusion requirement)-No significant coagulopathy (INR < 1.5, PT or aPTT > 1.5 times normal, platelet count > 50,000/mm^3^)-Hemodynamic stability (mean arterial pressure > 65 mmHg without vasopressor support)-No high bleeding-risk conditions:-Large intracranial hemorrhage (>10 mL volume or >1 cm thickness)-High-grade solid organ injury with active extravasation or expansion-Recent surgery (<24 h) with active oozing-Uncorrected severe coagulopathy

The patients were monitored daily for signs and symptoms of VTE and complications of VTE prophylaxis. Additionally, weekly surveillance of the lower extremities was conducted using duplex sonography.

All patients were monitored daily by the attending surgical team for clinical signs and symptoms of VTE, including:-Lower extremity symptoms: unilateral leg pain, swelling, warmth, erythema, or a palpable cord-Pulmonary symptoms: sudden dyspnea, chest pain, hemoptysis, tachypnea, or hypoxemia

Ultrasound Surveillance:

High-risk patients underwent weekly screening duplex ultrasound of the bilateral lower extremities, performed by a certified surgeon under the supervision of a vascular surgeon.

Examinations included:

Technique: B-mode imaging with color Doppler and compression technique.

Veins examined:


-Proximal: Common femoral, femoral, and popliteal veins-Distal: Posterior tibial, peroneal, and anterior tibial veins (if symptomatic)-DVT criteria: Noncompressible vein segment, visible intraluminal thrombus, absent or diminished color flow, and absent augmentation with distal compression-Quality control: Positive findings were confirmed by a board-certified vascular surgeon; clinical correlation was required for treatment decisions-Additional ultrasound examinations were performed for any patient with clinical suspicion of DVT, regardless of scheduled surveillance.


PE Diagnosis:

Pulmonary embolism was investigated with CT pulmonary angiography in patients with clinical suspicion (sudden dyspnea, chest pain, hypoxemia), unexplained tachycardia or hypotension, and confirmed proximal DVT with respiratory symptoms.

Bleeding Surveillance:

Patients receiving pharmacologic prophylaxis were monitored daily for bleeding complications.

Bleeding was considered attributed to anticoagulation if it occurred at least 12 h after the first dose and resolved within 48 h of discontinuation.

Clinical assessment for signs of bleeding

-Daily hemoglobin measurement-Platelet count every 3 days-For patients with traumatic brain injury: Repeat head CT within 24–48 h after starting anticoagulation, then as clinically indicated

All examinations were performed as part of routine clinical care, not specifically for research purposes.

### 2.4. Outcomes

The primary outcome of this study was the rate of venous thromboembolism in trauma patients. The secondary outcome was bleeding complications associated with the pharmacologic VTE prophylaxis method.

Patients’ data, which included demographics, comorbidities, injury mechanisms, vital signs, Abbreviated Injury Scale (AIS), concomitant injuries, Greenfield risk assessment profiles score, the prevalence of venous thromboembolism, and bleeding complications of VTE prophylaxis, were collected.

Major bleeding was defined according to the International Society on Thrombosis and Haemostasis [12] as fatal bleeding, symptomatic bleeding in a critical area or organ, such as intracranial, intraspinal, intraocular, retroperitoneal, intra-articular, or pericardial, or intramuscular with compartment syndrome, bleeding that caused a fall in hemoglobin levels of 1.24 mmol/L (20 g/L or greater) or more, or led to a transfusion of 2 units or more of whole blood or red cells.

Minor bleeding was defined as clinically overt bleeding that did not meet major criteria but required medical attention, including gastrointestinal hemorrhage (hematemesis, melena, or hematochezia requiring endoscopy or transfusion < 2 units), hematuria (gross hematuria requiring investigation or intervention), hematoma at the injury site requiring drainage or causing concern, and persistent wound bleeding requiring intervention beyond routine care.

### 2.5. Data Analysis

Statistical analysis was performed using IBM SPSS Statistics Version 28. Continuous variables were reported as means with standard deviations. Continuous data were compared using Student’s *t*-test or Mann–Whitney U-test. Categorical data were compared using a Chi-squared or Fisher’s exact test. Significance was set for *p* values < 0.05.

## 3. Results

A total of 4199 patients were admitted to the trauma surgery department during the study period, and 580 cases were eligible for the study. There were 430 male patients (74.1%) and 150 female patients (25.9%). Of the 580 cases, 310 (53.4%) were classified as low-risk, while 270 (46.6%) were high-risk. Both groups were comparable in terms of age and gender (Table 1). Pharmaco-mechanical VTE prophylaxis was administered to 30.4 percent (82 cases) of the high-risk group and 1 percent (3 cases) of the low-risk group (Figure 1).

Among high-risk patients, 188 (69.6%) received SCDs only. The main contraindications to pharmacologic prophylaxis included severe traumatic brain injury (AIS head > 2) in 105 patients (55.9% of those not on anticoagulation), high-grade solid organ injury in 54 patients (28.7%), and active coagulopathy in 60 patients (31.9%). Many patients had multiple overlapping contraindications.

The patients receiving pharmacologic prophylaxis included 63 individuals who received new prescriptions—comprising 60 for enoxaparin and 3 for heparin—while 19 patients continued their home anticoagulant regimens for other indications, with 11 on warfarin and 8 on direct oral anticoagulants. The median time to the initiation of prophylaxis was 3 days (Interquartile Range 2–5, Range 1–10 days). A total of 35 patients (55.6%) received early prophylaxis within 3 days, whereas 28 patients (44.4%) experienced delayed initiation exceeding 3 days, due to the development of evolving contraindications.

A total of 8 cases of VTE were identified. It was observed that all cases of VTE occurred in individuals who were at high risk. The overall rate of VTE was 1.4 percent (8 cases), DVT was 1.2 percent (7 cases), and PE was 0.2 percent (1 case). There was VTE in 3 percent of cases, DVT in 2.6 percent, and PE in 0.4 percent of high-risk cases (Table 2). The median time from trauma to VTE diagnosis was 14 days (IQR 7–21, range 5–24 days). Among patients receiving prophylaxis, VTE occurred at a median of 10 days after initiation (range 1–17 days). Detailed characteristics of all VTE cases were presented in Table 3. There was VTE in 5 cases while receiving anticoagulant, accounting for 6.1 percent of the pharmacological prophylaxis for high-risk (Table 3).

Table 4 presents both crude (univariate) and adjusted (multivariable) odds ratios for VTE risk factors. After multivariable adjustment, three variables independently predicted VTE occurrence. The RAP score and abdominal injury severity were the independent risk factors.

Cox regression analysis of the RAP score as a continuous variable demonstrated a significant association between increasing RAP score and VTE risk (HR 1.09, 95% CI 1.04–1.14, *p* < 0.001) (Figure 2).

Kaplan–Meier analysis showed that patients with abdominal injury had a significantly lower VTE-free survival rate than those without abdominal injury. The difference between the groups was statistically significant (log-rank test, *p* < 0.001).

In the Cox proportional hazards model, the presence of abdominal injury was associated with a substantially increased risk of VTE (hazard ratio [HR] 2.45, 95% confidence interval [CI] 1.68–3.59, *p* < 0.001), as illustrated in the forest plot (Figure 3).

The pharmacological prophylaxis-related major bleeding complications occurred in 4 cases (4.9%), and minor bleeding complications occurred in 8 cases (9.8%) out of 82 cases in the high-risk VTE group.

There was no pharmacological prophylaxis-related bleeding complication in the low-risk VTE group.

## 4. Discussion

The Western Trauma Association (WTA) recommended a decision algorithm to prevent VTE that involved early use of both mechanical and pharmacologic methods [9]. However, the main challenge was initiating pharmacologic prophylaxis since it increased the risk of bleeding for the patient.

The literature and clinical practice suggest delays in pharmacologic prophylaxis for patients with an active bleed (a hemoglobin drop of greater than 2 g/dL within 12 h or ongoing blood transfusion), coagulopathy (an elevated prothrombin time of more than 3 s above control or a platelet count of less than 50,000 per cubic millimeter), presence of hemodynamic instability, solid organ injury, traumatic brain injury (TBI), or spinal trauma [4].

A common cause of injury in Thailand is road traffic accidents. Most of the patients admitted to our center had sustained blunt trauma with injuries at multiple sites.

According to the study, pharmaco-mechanical was used in 30.4% of high-risk VTE patients. However, pharmacologic prophylaxis faced a major limitation due to the risk of bleeding. The high-risk group for VTE, including patients with head injuries and an AIS score > 2, accounted for 38.9% of cases, and the abdomen AIS score > 2 accounted for 20%. The concern for an increase in intracranial and intra-abdominal hemorrhage was a common reason for delaying the initiation of pharmacologic prophylaxis.

The rates of VTE varied among ethnic and genetic groups [13], with African Americans having higher rates than Caucasians and other groups [14]. Asians and Pacific islanders had the lowest rates [15]. The reported incidence of venous thromboembolism (VTE) after trauma ranged from 1.8% to 58% [1,7,16,17]. Our incidence rate of VTE was 1.4%, which was lower than what was reported in trauma literature.

All 8 cases of VTE events occurred in high-risk patients. Most cases were associated with abdominal and intracranial lesions. Five cases of VTE developed despite pharmacological prophylaxis, and all had a delayed initiation of anticoagulant prophylaxis due to bleeding at the injury site.

This document presents both crude (univariate) and adjusted (multivariable) odds ratios concerning VTE risk factors. In the univariate analysis, several factors demonstrated significant associations with the occurrence of VTE: RAP score (crude OR 1.671 per point, 95% CI: 1.313–2.126, *p* < 0.001), abdominal injury severity (crude OR 1.678 per AIS level, 95% CI: 1.197–2.354, *p* = 0.003), and receipt of pharmacologic prophylaxis (crude OR 8.312, 95% CI: 1.857–37.22, *p* = 0.006). Following multivariable adjustment, two variables independently predicted the occurrence of VTE. The RAP score remained the predictor (adjusted OR 1.493 per point, 95% CI: 1.123–1.986, *p* = 0.006), indicating that each one-point increase in the RAP score conferred a 49.3% increased likelihood of VTE, independent of other factors. These findings supported the incorporation of RAP score into routine VTE risk assessment protocols in trauma care. Moreover, abdominal injury severity demonstrated independent prognostic value beyond the RAP score (adjusted OR 1.458 per AIS level, 95% CI: 1.001–2.125, *p* = 0.049), implying that severe abdominal trauma contributed to VTE risk through mechanisms not entirely encompassed by the RAP scoring system. Kaplan–Meier survival analysis demonstrated significantly lower VTE-free survival in patients with abdominal injury.

A retrospective review of hospitalizations for adult blunt trauma reported that VTE events occurred at a median of 6 days (IQR 3–11), with 7.3% occurring within 1 day of admission [18]. A delay of pharmacologic prophylaxis for more than 4 days increased the risk of VTE by three times in major trauma patients (risk ratio, 3.0; 95% CI [1.4–6.5]) [19].

The incidence of venous thromboembolism in cases of traumatic brain injury was reported to be as high as 20–25% in the absence of medical prophylaxis or when it was delayed [20,21]. According to the literature, administering early prophylaxis was linked to lower rates of both pulmonary embolism (with an odds ratio of 0.48 and a 95% confidence interval ranging from 0.25 to 0.91) and deep vein thrombosis (with an odds ratio of 0.51 and a 95% confidence interval ranging from 0.36 to 0.72). This was achieved without increasing the risk of late neurosurgical intervention or death [22].

A WTA [9] recommended that patients with cerebral contusion, localized petechial hemorrhages, or diffuse axonal damage could safely receive pharmacologic prophylaxis at an early stage. The Kandahar Airfield neurosurgery service proposed initiating pharmacologic prophylaxis 24 h after injury for patients with a stable CT, which was safe and showed similar progression rates regardless of prophylaxis [23].

Our study found that four patients experienced progression of their intracranial hemorrhage lesions after starting pharmacologic prophylaxis medication. However, no surgical procedures were performed after the medication was stopped. It is important to note that bleeding events occurred in patients with large lesions of traumatic subdural hematoma before they began administering the anticoagulant medication. For patients with large intracranial hemorrhage, follow-up CT should have been performed after introducing pharmacologic prophylaxis as a predictor for further progression.

The optimal timing for the initiation of VTE prophylaxis following an abdominal solid organ injury remains a topic of ongoing discussion. A prospective study was conducted to examine the nonoperative management of blunt solid organ injury [24]. The results indicated that prophylaxis could be safely initiated within 48 h without increasing the need for post-prophylaxis transfusion or failure rates of nonoperative management. However, the study enrolled a few patients with high-grade injuries. In the absence of active bleeding, early initiation of VTE prophylaxis was recommended.

In cases of lower extremity vascular injury, there was a risk of thrombus formation due to direct intimal injury and immobilization. Studies have shown that the rate of VTE after complex lower extremity injuries ranged from 12.2 to 40.3% [25,26,27]. The rate of VTE after compound lower-limb fracture surgery without pharmacological prophylaxis was 46% [28]. Even though pharmacologic prophylaxis was started, VTE still occurred. Therefore, it was suggested not to delay VTE prophylaxis in these patients and to perform routine clinical surveillance and duplex sonography after admission.

The potential sites for minor bleeding complications included gastrointestinal hemorrhage, hematuria, hematoma at the injury site (closed fracture), and hemorrhage at the injury wound. These clinical conditions should have been monitored when pharmacologic prophylaxis was initiated.

Mechanical prophylaxis was safe and effective for low-risk VTE cases, regardless of concurrent pharmacologic prophylaxis. No VTE event was reported in this group in the study. Selecting low-risk patients with short hospital stays may not have required pharmacologic prophylaxis after minor trauma, while patients who were confined to bed did require pharmacologic VTE prophylaxis [9].

### Limitations

It is important to note that this study has some limitations, which are to be expected in a retrospective study. A future prospective study is required to gain more insights and a further understanding.

## 5. Conclusions

The Greenfield RAP score successfully identified all VTE cases as high-risk, with no VTE events in the low-risk group, supporting its utility for risk stratification in our population. The risk of pharmacologic prophylaxis-associated bleeding was weighed against the risk of VTE. The high risk of bleeding cases required close monitoring of this complication.

## Figures and Tables

**Figure 1 ijerph-23-00059-f001:**
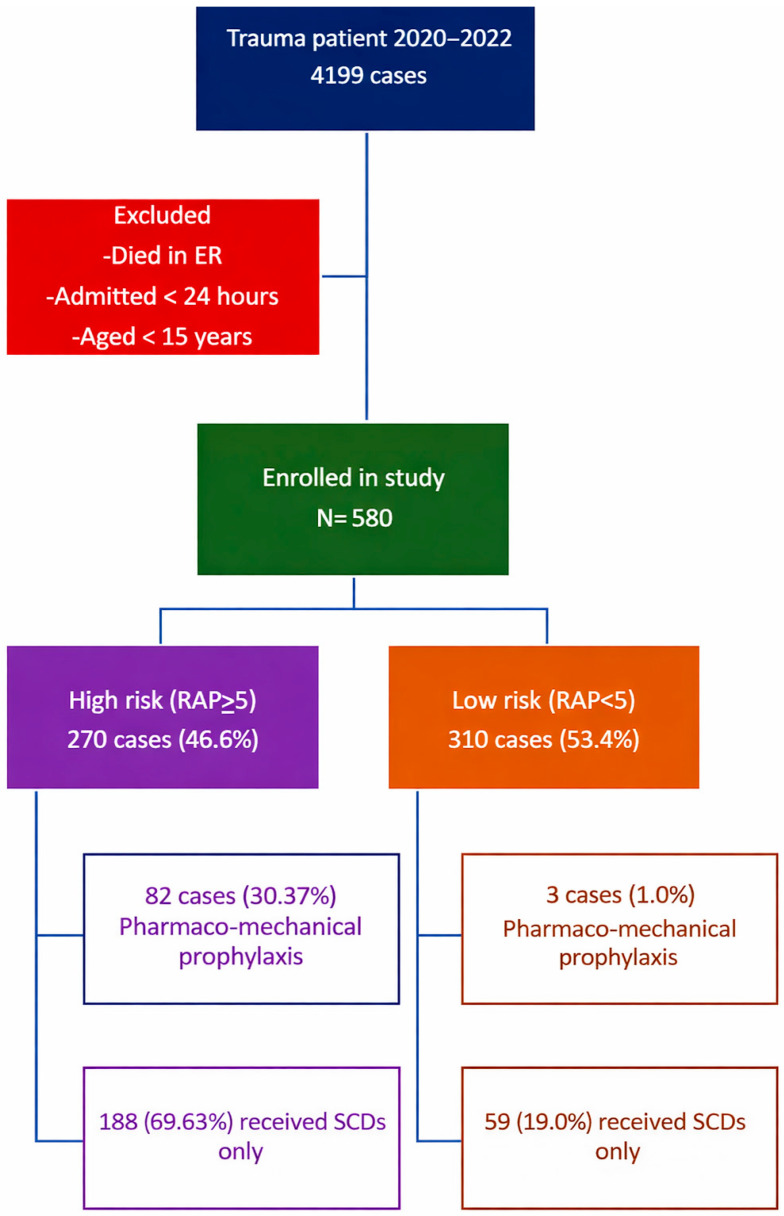
Flow diagram of cohort selection and prophylaxis allocation.

**Figure 2 ijerph-23-00059-f002:**
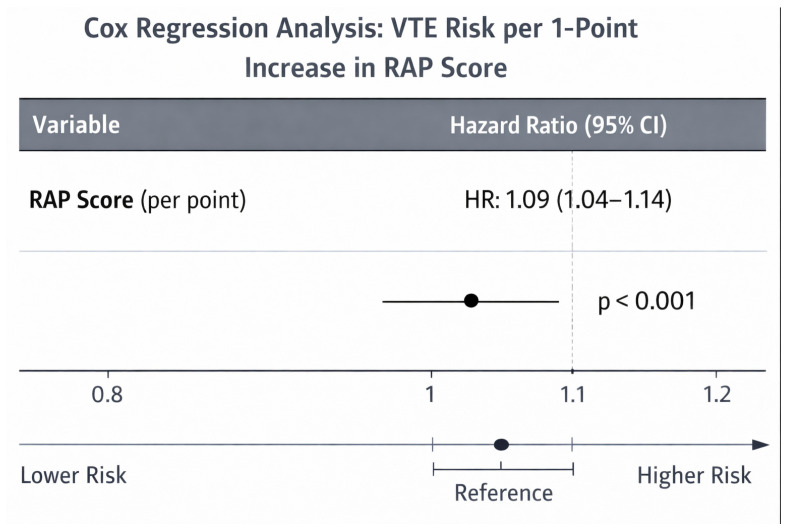
Hazard Ratio for VTE per 1 point increase in RAP score.

**Figure 3 ijerph-23-00059-f003:**
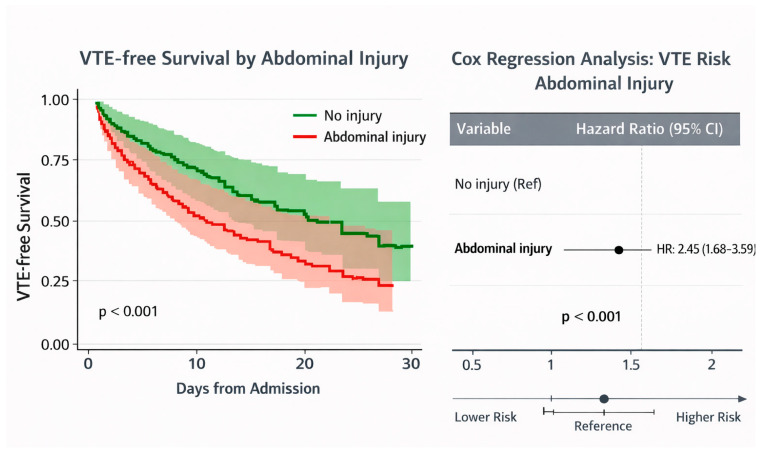
VTE-Free survival and VTE risk by abdominal injury.

**Table 1 ijerph-23-00059-t001:** Clinical characteristics of study population.

Variables	High Risk (*n* = 270)	Low Risk (*n* = 310)	*p*-Value	OR (95% CI)
Age (year), mean (SD)	53.57 (19.89)	42.86 (18.43)	<0.001 *	1.030 (1.019–1.041)
Gender, male, *n* (%)	195 (72.22)	236 (76.13)	0.283	0.813 (0.556–1.189)
ISS score, mean (SD)	19.01 (11.51)	10.73 (7.34)	<0.001 *	1.090 (1.069–1.112)
Greenfield Risk Assessment Profiles				
Risk assessment score, mean (SD)	8.88 (2.32)	3.11 (1.60)	<0.001 *	MD 5.77 (5.33–6.21) ‡ §
Underlying conditions				
Obesity (BMI > 30 kg/m^2^), *n* (%)	21 (7.78)	12 (3.87)	0.042 *	2.088 (1.014–4.298)
Malignancy, *n* (%)	14 (5.18)	3 (0.97)	0.003 *	5.619 (1.588–19.88)
Abnormal coagulation factors at admission, *n* (%)	60 (22.22)	7 (2,26)	<0.001 *	12.32 (5.518–27.50)
History of thromboembolism	52 (19.26)	4 (1.29)	<0.001 *	17.97 (6.376–50.63)
Iatrogenic factors				
Femoral central venous catheter >24 h, *n* (%)	4 (1.48)	0 (0)	0.046 *	∞ (0.87–∞) ¶
Four or more transfusions in 24 h, *n* (%)	23 (8.51)	1 (0.32)	<0.001 *	28.74 (3.841–215.0)
Surgical procedure > 2 h, *n* (%)	161 (59.62)	38 (12.26)	<0.001 *	10.80 (7.034–16.58)
Repair or ligation of major vascular injury (any named vessel), *n* (%)	40 (14.81)	4 (1.29)	<0.001 *	13.32 (4.674–37.97)
Injury-related factors				
AIS chest > 2, *n* (%)	78 (28.29)	34 (10.97)	<0.001 *	3.280 (2.098–5.130)
AIS abdomen > 2, *n* (%)	54 (20.00)	15 (4.84)	<0.001 *	4.899 (2.689–8.927)
AIS head > 2, *n* (%)	105 (38.89)	85 (27.42)	0.010 *	1.685 (1.187–2.391)
GCS score < 8 for >4 h, *n* (%)	26 (9.63)	4 (1.29)	<0.001 *	8.134 (2.788–23.73)
Complex lower extremity fracture, *n* (%)	67 (24.81)	13 (4.19)	<0.001 *	7.484 (4.020–13.93)
Pelvic fracture, *n* (%)	22 (8.15)	1 (0.32)	<0.001 *	27.28 (3.618–205.7)
AGE, *n* (%)			<0.001 *	
40–59	101 (37.41)	97 (31.29)		2.322 (1.486–3.629)
60–74	56 (20.74)	54 (17.42)		2.314 (1.404–3.814)
>75	49 (18.15)	13 (4.19)		8.410 (4.207–16.81)

* *p* < 0.05, statistically significant. Odds ratio (95% CI) for high-risk vs. low-risk from univariate logistic regression. ‡ Mean difference (95% CI) from independent *t*-test. § RAP score used to define risk groups; OR not calculated. ¶ Infinite OR because all events in one group; CI from exact logistic regression.

**Table 2 ijerph-23-00059-t002:** Venous Thromboembolism and Bleeding Complications.

Variables	High Risk (*n* = 270)	Low Risk (*n* = 310)	*p*-Value
Venous thromboembolism, *n* (%)	8 (2.96)	0 (0)	0.002 *
Deep vein thrombosis	7 (2.59)		
Pulmonary embolism	1 (0.37)		
Major bleeding complication, *n* (%)		0 (0)	0.026 *
Intracranial hemorrhage	4 (1.48)		
Minor bleeding complication, *n* (%)		0 (0)	0.037 *
Gastrointestinal hemorrhage	3 (1.11)		
Hematuria	3 (1.11)		
Hematoma	1 (0.37)		
Bleeding per wound	1 (0.37)		

* *p* < 0.05, statistically significant.

**Table 3 ijerph-23-00059-t003:** A summary of cases involving venous thromboembolism.

Case	Age	Gender	Injury	Risk Assessment Profile Score	Start Date of Anticoagulant After Trauma	VTE Event
1	67	female	blunt traumatic jejunal injury grade, laceration wound at left knee, cerebral concussion	10	7	DVT
2	33	male	occipital condyle displacement, left SDH, complex maxillofacial injury	10	4	DVT
3	64	male	multiple rib fracture with left pneumothorax, pancreaticoduodenal injury, left subdural hemorrhage, C3–C4 fracture	10	-	DVT
4	34	male	posterior knee dislocation with popliteal artery injury, subarachnoid hemorrhage with subdural hemorrhage, C3–C5 spinous process fracture	10	7	DVT
5	26	male	right 6–10th rib fracture with hemothorax, liver laceration, right kidney Injury, left femoral shaft fracture and closed fracture subtrochanteric of right femur, sacral fracture (S1–S2), right Iliac crest fracture, traumatic brachial plexus injury lower arm type, traumatic SDH at falx cerebri, rhabdomyolysis	15	7	DVT
6	68	female	bilateral hemothorax, duodenal injury, fracture sacrum, iliac bone, posterior wall of right acetabulum, left pubic tubercle, right superior and inferior pubic rami	15	-	DVT
7	40	female	liver laceration, C7 spinal process fracture, comminuted fracture of greater wing of right sphenoid bone, right sphenozygomatic suture, right maxillary sinus, and right sphenoid sinus, closed fracture left distal end radius	10	-	DVT
8	77	male	open fracture right tibia, Cerebral concussion, close fracture right ulnar	11	5	PE

**Table 4 ijerph-23-00059-t004:** Univariate and Multivariable Logistic Regression Analysis for VTE Risk Factors.

Variable	Crude OR (Univariate)	*p*-Value	95% CI	Adjusted OR (Multivariate)	*p*-Value	95% CI
Primary multivariable model (*n* = 580, 8 VTE events)						
RAP score (per point increase)	1.671	1.313–2.126	<0.001 *	1.493	1.123–1.986	0.006 *
AIS abdomen (per severity level)	1.678	1.197–2.354	0.003 *	1.458	1.001–2.125	0.049 *
Pharmacologic prophylaxis	8.312	1.857–37.22	0.006 *	2.797	0.606–12.906	0.188

* *p* < 0.05, statistically significant.

## Data Availability

The original contributions presented in this study are included in the article. Further inquiries can be directed to the corresponding author.

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
