# Peer review of "Venous Thromboembolism Risk Assessment and Prophylaxis in Trauma Patients"

_ijerph, 2025, doi:10.3390/ijerph23010059_

Round 1
Reviewer 1 Report
Comments and Suggestions for Authors
Dear Authors! I appreciate your efforts in conducting your study. I have some remarks.
Major remarks
- Lines 85-86. I did not quite catch how this sample size of 207 cases had been calculated. The way used for sample size calculation must be presented in the Statistical analysis section. Please, present in details how the sample size had been calculated.
- Please, describe prophylaxis protocol more clearly. As of now it looks as you prescribed IPC and anticoagulants for those with score 5 > as well as for those with score <5. Who did receive no prophylaxis at all? Who did receive either IPC alone or anticoagulants alone?
- Please, specify which anticoagulants were used for VTE prevention and in which dosages?
- Please, specify the exact regimen of IPC that was used for VTE prevention? How many hours a day? How many days?
- Please, present the cohort selection in a flow-chart. How many patients were admitted, how many patients were excluded and the reasons for that with exact numbers of excluded subjects, how many patients in low and high risk groups received different prophylaxis tools - IPC+ACT, IPC alone, ACT alone, no prophylaxis.
- Please, present reasons for not prescribing VTE prophylaxis in high risk patients. Only 30.4% high risk patients did receive prophylaxis in presented cohort. Why others did not receive? What contraindications did you register and how often you registered every contraindication?
- Please, explain why low risk patients received IPC and ACT for prophylaxis.
- Conclusions are not correct. Your data don’t confirm VTE reduction by implementing risk assessment protocol. To confirm this you had to compare VTE rates in two cohorts - with a protocol used and without. What did you find is the VTE rate in real life setting in a specific cohort managed with this protocol. You did not also confirm that Greenfield score is a reliable tool. This statement also needs comparison of this score with another tool. Please, limit your conclusions to what you found - VTE and bleedings in a specific cohort.
Minor remarks
- Reference 1 is nor fully relevant. These are guidelines that did not study morbidity and mortality in trauma patients. Please, refer to a study or review that discussed this subject.
- Reference 3 is fully irrelevant. This study did not investigate all that you mentioned in the lines 50-53.
- Lines 95-97. Please, create a section “2.3. Patients examination”. Please, present here in details who did perform clinical examination, a nurse or physician? Which signs and symptoms were monitored? Which kind of ultrasound was performed? Whole leg or POCUS? Did you monitor only proximal veins or calf veins were examined as well? Did DUS was performed by someone from investigators team or you just extracted data from routinely performed examinations?
- Line 99. It is better to use rate rather than prevalence.
- Line 100. No need to say “after using ... the tool”. The end-point is the rate of VTE.
- How did you were going to distinguish bleedings associated with anticoagulants from bleedings of another reasons? Please, explain or just skip this definition.
- Please, specify in Materials and Methods which kinds of bleedings were registered. Only major? Or all bleedings, including clinically relevant non-major and minor ones?
- Lines 102-105. Please, move this text to new section 2.3 (see above).
- Table 2 has to be re-titled. VTE is no complication of prophylaxis. This table represents VTE rates.
- Lines 141-145 has to be deleted. Those statements are ok for background but not for Discussion.
Author Response
Thank you for your valuable feedback. We have thoroughly considered each point and made substantial revisions to our manuscript. Here is our detailed response to each concern:
The attached file highlights the change in paper as reviewer 1 recommended in blue.
- Lines 85-86. I did not quite catch how this sample size of 207 cases had been calculated. The way used for sample size calculation must be presented in the Statistical analysis section. Please, present in details how the sample size had been calculated.
We calculated the sample size based on an expected VTE prevalence of 5% in the Thai trauma population [ref 11]. Using the formula for a single-proportion study: n = Z²×p(1-p) / d², where Z=1.96, p=0.05, and d=0.02. The minimum required was 207 patients. However, to ensure adequate power for subgroup analyses and account for potential exclusions, we included all eligible trauma patients during the study period (N=580), which exceeded our minimum requirement and enhanced study precision.
- Please, describe prophylaxis protocol more clearly. As of now it looks as you prescribed IPC and anticoagulants for those with score 5 > as well as for those with score <5. Who did receive no prophylaxis at all? Who did receive either IPC alone or anticoagulants alone?
Our institution implemented a standardized VTE prophylaxis protocol based on the Greenfield Risk Assessment Profile (RAP) score [10]. All trauma patients admitted for more than 24 hours underwent risk stratification on admission.
Mechanical Prophylaxis was administered to all bed-bound patients through the use of bilateral sequential compression devices (SCDs), which were employed for 18-20 hours daily until the patients became ambulatory or were discharged. The devices were temporarily removed solely for ambulation, physical therapy, bathing, or patient comfort, with a maximum duration of 2 hours.
Pharmacologic prophylaxis included enoxaparin 40 mg subcutaneously once daily, reduced to 30 mg if creatinine clearance (CrCl) was below 30 mL/min. In patients with severe renal failure (CrCl < 15), when enoxaparin was contraindicated, unfractionated heparin 5,000 units was administered subcutaneously every 8 hours. Alternatively, patients could continue their home anticoagulants for other reasons.
Prophylaxis was initiated when all of the following criteria were met
- No active bleeding (stable hemoglobin for >12 hours, no ongoing transfusion requirement)
- No significant coagulopathy (INR < 1.5, PT or aPTT > 1.5 times normal, platelet count > 50,000/mm³)
- Hemodynamic stability (mean arterial pressure > 65 mmHg without vasopressor support)
- No high bleeding-risk conditions:
- Large intracranial hemorrhage (>10 mL volume or >1 cm thickness)
- High-grade solid organ injury with active extravasation or expansion
- Recent surgery (<24 hours) with active oozing
- Uncorrected severe coagulopathy
- Please, specify which anticoagulants were used for VTE prevention and in which dosages? Pharmacologic prophylaxis included enoxaparin 40 mg subcutaneously once daily, reduced to 30 mg if creatinine clearance (CrCl) was below 30 mL/min. In patients with severe renal failure (CrCl < 15), when enoxaparin was contraindicated, unfractionated heparin 5,000 units was administered subcutaneously every 8 hours. Alternatively, patients could continue their home anticoagulants for other reasons.
4. Please, specify the exact regimen of IPC that was used for VTE prevention? How many hours a day? How many days?
Mechanical Prophylaxis was administered to all bed-bound patients through the use of bilateral sequential compression devices (SCDs), which were employed for 18-20 hours daily until the patients became ambulatory or were discharged. The devices were temporarily removed solely for ambulation, physical therapy, bathing, or patient comfort, with a maximum duration of 2 hours.
The SCDs delivered graded compression (45 mmHg at the ankle, decreasing proximally). Active DVT in the limb was a contraindication. The pressure, cycle, and duration of compression were adjusted for patients with peripheral arterial disease. The device was modified and partially applied only to areas not affected by acute fractures or severe open wounds.
5. Please, present the cohort selection in a flow-chart. How many patients were admitted, how many patients were excluded and the reasons for that with exact numbers of excluded subjects, how many patients in low and high risk groups received different prophylaxis tools - IPC+ACT, IPC alone, ACT alone, no prophylaxis.
a flow-chart was added.
- Please, present reasons for not prescribing VTE prophylaxis in high risk patients. Only 30.4% high risk patients did receive prophylaxis in presented cohort. Why others did others not receive? What contraindications did you register and how often you registered every contraindication?
Among high-risk patients, 188 (69.6%) received SCDs only. The main contraindications to pharmacologic prophylaxis included severe traumatic brain injury (AIS head >2) in 105 patients (55.9% of those not on anticoagulation), high-grade solid organ injury in 54 patients (28.7%), and active coagulopathy in 60 patients (31.9%). Many patients had multiple overlapping contraindications.
- Please, explain why low risk patients received IPC and ACT for prophylaxis.
These patients have prolonged hospital stays due to their medical condition, the team has implemented anticoagulant therapy for VTE prophylaxis.
- Conclusions are not correct. Your data don’t confirm VTE reduction by implementing risk assessment protocol. To confirm this you had to compare VTE rates in two cohorts - with a protocol used and without. What did you find is the VTE rate in real life setting in a specific cohort managed with this protocol. You did not also confirm that Greenfield score is a reliable tool. This statement also needs comparison of this score with another tool. Please, limit your conclusions to what you found - VTE and bleedings in a specific cohort.
We completely agree and have substantially revised our conclusions to accurately reflect our findings
Revised conclusions:
The Greenfield RAP score successfully identified all VTE cases as high-risk with no VTE events in the low-risk group, supporting its utility for risk stratification in our population.
- Reference 1 is nor fully relevant. These are guidelines that did not study morbidity and mortality in trauma patients. Please, refer to a study or review that discussed this subject.
We have replaced Reference 1 with more appropriate citations.
- Reference 3 is fully irrelevant. This study did not investigate all that you mentioned in the lines 50-53.
We have replaced Reference 3 with more appropriate citations.
- Lines 95-97. Please, create a section “2.3. Patients examination”. Please, present here in details who did perform clinical examination, a nurse or physician? Which signs and symptoms were monitored? Which kind of ultrasound was performed? Whole leg or POCUS? Did you monitor only proximal veins or calf veins were examined as well? Did DUS was performed by someone from investigators team or you just extracted data from routinely performed examinations?
All patients were monitored daily by the attending surgical team for clinical signs and symptoms of VTE, including:
Lower extremity symptoms: unilateral leg pain, swelling, warmth, erythema, or a palpable cord
Pulmonary symptoms: sudden dyspnea, chest pain, hemoptysis, tachypnea, or hypoxemia
Ultrasound Surveillance: High-risk patients underwent weekly screening duplex ultrasound of the bilateral lower extremities, performed by a certified surgeon under the supervision of a vascular surgeon.
Examinations included:
Technique: B-mode imaging with color Doppler and compression technique
Veins examined:
Proximal: Common femoral, superficial femoral, and popliteal veins
Distal: Posterior tibial, peroneal, and anterior tibial veins (if symptomatic)
DVT criteria: Noncompressible vein segment, visible intraluminal thrombus, absent or diminished color flow, and absent augmentation with distal compression
Quality control: Positive findings were confirmed by a board-certified vascular surgeon; clinical correlation was required for treatment decisions
Additional ultrasound examinations were performed for any patient with clinical suspicion of DVT, regardless of scheduled surveillance.
PE Diagnosis:
Pulmonary embolism was investigated with CT pulmonary angiography in patients with clinical suspicion (sudden dyspnea, chest pain, hypoxemia), unexplained tachycardia or hypotension, confirmed proximal DVT with respiratory symptoms
Bleeding Surveillance:
Patients receiving pharmacologic prophylaxis were monitored daily for bleeding complications:
Clinical assessment for signs of bleeding
Daily hemoglobin measurement
Platelet count every 3 days
For traumatic brain injury patients: Repeat head CT within 24-48 hours after starting anticoagulation, then as clinically indicated
All examinations were performed as part of routine clinical care, not specifically for research purposes.
- Line 99. It is better to use rate rather than prevalence.
The word was revised.
- Line 100. No need to say “after using ... the tool”. The end-point is the rate of VTE.
The phase was removed.
- How did you were going to distinguish bleedings associated with anticoagulants from bleedings of another reasons? Please, explain or just skip this definition.
Bleeding was attributed to anticoagulation if it occurred ≥12 hours after first dose and resolved within 48 hours of discontinuation."
7.Please, specify in Materials and Methods which kinds of bleedings were registered. Only major? Or all bleedings, including clinically relevant non-major and minor ones?
Major bleeding was defined according to the International Society on Thrombosis and Haemostasis [12] as fatal bleeding, symptomatic bleeding in a critical area or organ, such as intracranial, intraspinal, intraocular, retroperitoneal, intra-articular or pericardial, or intramuscular with compartment syndrome, bleeding causing a fall in hemoglobin levels of 1.24 mmol/L (20 g/L or greater) or more or leading to a transfusion of 2 units or more of whole blood or red cells.
Minor bleeding was defined as clinically overt bleeding that did not meet major criteria but required medical attention, including gastrointestinal hemorrhage (hematemesis, melena, or hematochezia requiring endoscopy or transfusion <2 units), hematuria (gross hematuria requiring investigation or intervention), hematoma at the injury site requiring drainage or causing concern, and persistent wound bleeding requiring intervention beyond routine care.
- Lines 102-105. Please, move this text to new section 2.3 (see above).
Revised title Venous Thromboembolism and Bleeding Complications
- Lines 141-145 has to be deleted. Those statements are ok for background but not for Discussion. : The sentenses were deleted.
Reviewer 2 Report
Comments and Suggestions for Authors
Please elaborate more on the Greenfield risk assessment profile score in the Introduction .
Please explain which anticoagulants were used.
Please consider adding confidence intervals in Table 1.
Consider the use of multivariable logistic regression to minimize confounding.
The statistical analysis could be better- The use of multivariate models is recommended to limit confounding factors. Kaplan Meier curves could be used
The authors could add more information, such as the timing of DVT diagnosis after initiation of prophylaxis and trauma event, the timing of initiation of prophylaxis, choice of anticoagulants.
There are numerous grammatical errors, redundancies in the discussion, and sections needing clearer organization. The manuscript would benefit from thorough editing before acceptance.
Consider adding information such as the timing of DVT diagnosis after initiation of prophylaxis and trauma event, timing of initiation of prophylaxis
Consider correcting grammatical errors and redundancies in the discussion.
Author Response
Thank you for your valuable feedback. We have thoroughly considered each point and made substantial revisions to our manuscript. Here is our detailed response to each concern:
The attached file highlights the change in paper as reviewer 2 recommended in red.
Please elaborate more on the Greenfield risk assessment profile score in the Introduction .
The Greenfield Risk Assessment Profile, developed by Greenfield et al. in 1997 [10], is a validated tool for evaluating trauma patients. It assigns weighted points based on various risk factors:
- Age: 2 points (40-59 years), 3 points (60-74 years), 4 points (75 years and older)
- Underlying conditions: Obesity (BMI over 30), malignancy, history of VTE, abnormal coagulation studies (1 point each)
- Injury-related factors: Major surgery lasting more than 2 hours, blood transfusion of 4 or more units in 24 hours, femoral central line in place for more than 24 hours, major vascular injury (1 point each)
- Specific injuries: Pelvic fracture, complex lower extremity fracture, severe chest injury (AIS >2), severe abdominal injury (AIS >2), severe head injury (AIS >2), GCS less than 8 for more than 4 hours (1 point each)
Patients with scores >5 were classified as high-risk; those with scores ≤5 as low-risk.
Please explain which anticoagulants were used.
Pharmacologic prophylaxis included enoxaparin 40 mg subcutaneously once daily, reduced to 30 mg if creatinine clearance (CrCl) was below 30 mL/min. In patients with severe renal failure (CrCl < 15), when enoxaparin was contraindicated, unfractionated heparin 5,000 units was administered subcutaneously every 8 hours. Alternatively, patients could continue their home anticoagulants for other reasons.
Prophylaxis was initiated when all of the following criteria were met
- No active bleeding (stable hemoglobin for >12 hours, no ongoing transfusion requirement)
- No significant coagulopathy (INR < 1.5, PT or aPTT > 1.5 times normal, platelet count > 50,000/mm³)
- Hemodynamic stability (mean arterial pressure > 65 mmHg without vasopressor support)
- No high bleeding-risk conditions:
- Large intracranial hemorrhage (>10 mL volume or >1 cm thickness)
- High-grade solid organ injury with active extravasation or expansion
- Recent surgery (<24 hours) with active oozing
- Uncorrected severe coagulopathy
Please consider adding confidence intervals in Table 1.
The data were inserted in table 1.
Consider the use of multivariable logistic regression to minimize confounding.
The statistical analysis could be better- The use of multivariate models is recommended to limit confounding factors. Kaplan Meier curves could be used
The data were inserted into Table 4 and Kaplan–Meier.
The authors could add more information, such as the timing of DVT diagnosis after initiation of prophylaxis and trauma event, the timing of initiation of prophylaxis, choice of anticoagulants.
Among high-risk patients, 188 (69.6%) received SCDs only. The main contraindications to pharmacologic prophylaxis included severe traumatic brain injury (AIS head >2) in 105 patients (55.9% of those not on anticoagulation), high-grade solid organ injury in 54 patients (28.7%), and active coagulopathy in 60 patients (31.9%). Many patients had multiple overlapping contraindications.
The patients receiving pharmacologic prophylaxis included 63 individuals who received new prescriptions—comprising 60 for enoxaparin and 3 for heparin—while 19 patients continued their home anticoagulant regimens for other indications, with 11 on warfarin and 8 on direct oral anticoagulants. The median time to the initiation of prophylaxis was 3 days (Interquartile Range 2-5, Range 1-10 days). A total of 35 patients (55.6%) received early prophylaxis within 3 days, whereas 28 patients (44.4%) experienced delayed initiation exceeding 3 days, due to the development of evolving contraindications.
Consider adding information such as the timing of DVT diagnosis after initiation of prophylaxis and trauma event, timing of initiation of prophylaxis
The median time from trauma to VTE diagnosis was 14 days (IQR 7-21, range 5-24 days). Among patients receiving prophylaxis, VTE occurred a median of 10 days after initiation (range 1-17 days). Detailed characteristics of all VTE cases are presented in Table 3.

Round 2
Reviewer 1 Report
Comments and Suggestions for Authors
Dear Authors! Thank you for addressing my remarks.